# Novel Biomarkers to Distinguish between Type 3c and Type 2 Diabetes Mellitus by Untargeted Metabolomics

**DOI:** 10.3390/metabo10110423

**Published:** 2020-10-22

**Authors:** Cristina Jimenez-Luna, Ariadna Martin-Blazquez, Carmelo Dieguez-Castillo, Caridad Diaz, Jose Luis Martin-Ruiz, Olga Genilloud, Francisca Vicente, Jose Perez del Palacio, Jose Prados, Octavio Caba

**Affiliations:** 1Institute of Biopathology and Regenerative Medicine (IBIMER), Center of Biomedical Research (CIBM), University of Granada, 18012 Granada, Spain; crisjilu@ugr.es (C.J.-L.); jcprados@ugr.es (J.P.); ocaba@ugr.es (O.C.); 2Fundación MEDINA, Centro de Excelencia para la Investigación en Medicamentos Innovadores en Andalucía, 18012 Granada, Spain; ariadna.martin@medinaandalucia.es (A.M.-B.); caridad.diaz@medinaandalucia.es (C.D.); olga.genilloud@medinaandalucia.es (O.G.); francisca.vicente@medinaandalucia.es (F.V.); 3Department of Gastroenterology, San Cecilio University Hospital, 18012 Granada, Spain; carmelo89dc@gmail.com (C.D.-C.), jlmartin@ugr.es (J.L.M.-R.)

**Keywords:** metabolomics, untargeted LC-HRMS, diagnosis, pancreatogenic diabetes mellitus, type 2 diabetes mellitus, chronic pancreatitis, biomarker

## Abstract

Pancreatogenic diabetes mellitus (T3cDM) is a highly frequent complication of pancreatic disease, especially chronic pancreatitis, and it is often misdiagnosed as type 2 diabetes mellitus (T2DM). A correct diagnosis allows the appropriate treatment of these patients, improving their quality of life, and various technologies have been employed over recent years to search for specific biomarkers of each disease. The main aim of this metabolomic project was to find differential metabolites between T3cDM and T2DM. Reverse-phase liquid chromatography coupled to high-resolution mass spectrometry was performed in serum samples from patients with T3cDM and T2DM. Multivariate Principal Component and Partial Least Squares-Discriminant analyses were employed to evaluate between-group variations. Univariate and multivariate analyses were used to identify potential candidates and the area under the receiver-operating characteristic (ROC) curve was calculated to evaluate their diagnostic value. A panel of five differential metabolites obtained an area under the ROC curve of 0.946. In this study, we demonstrate the usefulness of untargeted metabolomics for the differential diagnosis between T3cDM and T2DM and propose a panel of five metabolites that appear altered in the comparison between patients with these diseases.

## 1. Introduction

Diabetes mellitus (DM) is characterized by the destruction or dysfunction of pancreatic β-cells, producing progressive hyperglycemia [1]. Because the mechanisms responsible for these changes likely differ between pancreatic disorders, it has been proposed that DM should be classified according to the underlying disorder [2]. Chronic pancreatitis (CP) is a pathological fibro-inflammatory syndrome accompanied by irreversible morphological changes, fibrosis, and impairment of both exocrine and endocrine functions of the pancreatic gland [3]. CP is frequently associated with pancreatogenic DM, recently designated as type 3c DM (T3cDM) [4].

It has been reported that around 30% of patients with CP develop T3cDM [5], which is often wrongly diagnosed as type 1 DM (T1DM) or especially type 2 DM (T2DM) [6]. In 2017, a study in the UK found that 559 of 31,789 adults newly diagnosed with DM had a history of pancreatic disease and that most of them (87.8%) were classified as T2DM [7]. The correct diagnosis of T3cDM and T2DM is important because they are both risk factors for pancreatic cancer (PC) [8,9] and because higher concentrations of hemoglobin A1c are observed in patients with T3cDM, who require insulin earlier in comparison to those with T2DM [10].

Metabolomics is an emerging and powerful discipline that provides an accurate and dynamic image of the phenotype of biological systems by studying endogenous and exogenous metabolites in cells, tissues, and biofluids. The aim of metabolic fingerprinting, through the non-targeted global analysis of tissues and biofluids, is to fingerprint and semiquantify metabolites and their changes, revealing information about the general metabolic state of the individual. Metabolomics analysis of human serum is highly useful, because serum is a rich source of potential biomarkers of the impairment of pathways in different disorders [11]. In this regard, liquid chromatography coupled to high resolution spectrometry (LC-HRMS) has become a reference analytical platform in the area of metabolomics due to its increased sensitivity and broad metabolic coverage [12,13]. This approach has been used to propose specific metabolite signatures for pancreatic ductal adenocarcinoma (PDAC) and T2DM and even for the stratification of diabetic patients and complications [14,15,16].

The objective of this study was to use untargeted metabolomics to identify potential biomarkers of T3cDM and T2DM, performing reverse-phase liquid chromatography (RPLC) coupled to high-resolution mass spectrometry (LC-HRMS) in serum samples from patients.

## 2. Results

### 2.1. LC-HRMS Analysis

A reverse-phase column was used to separate compounds with medium and low polarity. Total ion chromatograms (TIC) of T3cDM and T2DM representative samples (Figure 1) showed different retention times (RTs) for medium-polar metabolites (eluting in the range from 7 to 15 min), for highly polar metabolites (showing RT in the first 5 min), and for non-polar metabolites (observed in the late RT). The RTs obtained show that the chromatographic separation was successful, and a noteworthy between-group change was observed from minutes 10.5 to 12, corresponding to the RT of the majority of lipids.

### 2.2. Chemometric Analysis

Following alignment and filtering stages, 3998 metabolite features were retrieved from the raw data, comprising 866 monoisotopic peaks. From monoisotopic signals, 350 features were excluded for unacceptable variability (RSD > 30%); next, 279 features were differentially expressed between study samples (T3cDM and T2DM) and organic solvent (OS) samples and were rejected as contaminants. Therefore, 259 variables were considered in the principal component analysis (PCA). This multivariate analysis (Figure 2) displayed a tight grouping of QC samples, indicating that the differentiation between T3cDM and T2DM was mainly attributable to biological factors. Within this set of 259 variables, 26 met the criteria (pv ≤ 0.05 and 0.6 ≤ FC ≥ 1.5) for selection as candidate biomarkers.

### 2.3. Identification of Potential Biomarkers

The accurate mass value of each selected feature was interrogated against compound databases (Metlin, NIST, LipidMaps and Human Metabolome Database). As a result, several molecular formulas were obtained with a mass error below 10 ppm. Each molecular formula was ranked according to experimental accurate mass and isotopic profile in order to assign a tentative identification for each potential biomarker (Table 1).

By this means, five compounds were identified in four classes of lipids (amino acids, carnitines, bile acids, lysophospholipids and bilirubins). Furthermore, interpretation of experimental fragmentation spectra and spectral database searches yielded the following structural identifications:

Identification of *m*/*z* 188.0716 at 3.3 min was supported by a high matching rate between the experimental fragmentation spectrum and the spectrum stored in NIST14 DB corresponding to L-tryptophan (NIST#: 1075940 ID#: 146255).

For *m*/*z* 400.3408 at 9.9 min, the experimental MS/MS spectrum was highly similar to that corresponding to palmitoylcarnitine in NIST14 DB (NIST#: 1152359 ID#: 30873).

The experimental fragmentation spectrum of feature *m*/*z* 431.3140 at 11.85 min had a high matching rate with that corresponding to Cholest-4-en-26-oic acid, 7α-hydroxy-3-oxo (NIST#: 1,148,941 ID#: 27467) (Figure 3).

Characteristic fragment ions of lysophosphatidylethanolamine compound class, such as [M+H-141] = 385, corresponding to the neutral loss of the phosphoethanolamine group from the molecular ion, were observed for *m*/*z* 526.2931 at 10.6 min. Additionally, MS/MS interpretation of its corresponding [M+Na]^+^ adduct at *m*/*z* 548.2721 provided several fragment ions, including: [M+Na-43]^+^ = 505 (loss of aziridine), [M+Na-163] = 385 (loss of [(OH)_2_PO_2_(CH_2_)_2_PONH_2_Na]), [M+Na-141]^+^ = 407 (loss of [(OH)_3_PO(CH_2_)_2_PONH_2_] and [M+Na-61]^+^ = 487 (loss of [OH(CH_2_)_2_NH_2_]).

Identification of *m*/*z* 583.25 at 8.2 min was further supported by a high matching rate between the experimental fragmentation spectrum and the spectrum stored in NIST14 DB corresponding to biliverdin (NIST#: 1219743 ID#: 98197).

### 2.4. Biomarker Evaluation

The area under the receiver-operating characteristic (ROC) curve (AUC) calculated for each candidate marker ranged from 0.71 to 0.93, indicating the moderate clinical utility of each individual biomarker. However, in complex diseases such as cancer or diabetes, the combination of different individual markers in a multivariate model frequently delivers the necessary discriminative power, as in the present case. The AUC for the final model was 0.946 (95% CI 0.802–1), indicating its robust discriminative ability and supporting the potential of these metabolites as biomarkers (Figure 4A). Based on this model, only 2 out of 19 samples labeled as T2DM were wrongly classified as T3cDM, whereas 5 out of 21 samples labeled as T3cDM were misclassified as T2DM (Figure 4B).

## 3. Discussion

No standardized diagnostic criteria are currently available for T3cDM secondary to CP, hampering the differential diagnosis with other types of diabetes and leading to its frequent misdiagnosis as T2DM and an underestimation of its incidence [17]. The diagnosis of T3cDM requires the presence of exocrine pancreatic insufficiency (EPI), the absence of T1DM autoantibodies, and confirmation by pancreatic imaging findings [18]. Its misdiagnosis means that patients do not receive the appropriate therapy, which involves the treatment of EPI, the maintenance of adequate levels of fat-soluble vitamins (especially vitamin D), and the restoration of impaired fat hydrolysis [19]. Patients with T3cDM should undergo regular follow-ups with dietary assessment to prevent hypoglycemia, reduce hyperglycemia, avoid malnutrition, treat EPI, and reduce the risk of diabetes-related complications [20].

In the present study, we describe various metabolites that differed between T3cDM and T2DM groups. One of these, LysoPE, results from the partial hydrolysis of phosphatidylethanolamine, which removes one of the fatty acid groups, and it is a minor constituent of cell membranes. LysoPE(22:6) was previously found to be altered in the serum metabolome of patients with early-stage ovarian cancer versus healthy controls, and it was included in a 16-metabolite diagnostic model that identified early-stage ovarian cancer with 100% accuracy [21]. Concentrations of this metabolite were also reported to be significantly higher in patients with liver cirrhosis who developed hepatocellular carcinoma than in those who did not [22]. Ha et al. used LC-HRMS in their metabolic study of patients with newly diagnosed T2DM versus sex- and BMI-matched non-diabetic controls and reported fold-change values of 2.04 for lysoPE(22:6), which achieved an AUC of 0.746 to evaluate the risk of developing T2DM with a sensitivity of 0.808 and specificity of 0.667 [23].

As in the present study, the essential amino acid tryptophan and metabolites of its pathway have been widely associated with T2DM. In 2011, a prospective study of 2422 individuals with a 12-year follow-up reported a positive association between baseline tryptophan concentrations and subsequent incident T2DM [24]. In 2016, a prospective study of 213 participants with a 10-year follow-up found greater insulin resistance in patients with higher tryptophan concentrations, which were significantly increased in those who developed T2DM [25]. Finally, in 2018, the authors of a case-cohort study (*n* = 694) with a median follow-up of 3.8 years proposed baseline tryptophan levels as a risk marker for incident T2DM, with a hazard ratio of 1.29 [26].

Higher concentrations of palmitoylcarnitine were also observed in the present patients with T2DM, and this carnitine has been widely proposed as an early biomarker for this disease, even before the onset of insulin resistance. Alterations in serum acylcarnitine profiles were initially associated with worse glucose tolerance, mediated by mitochondrial dysregulation and incomplete long-chain fatty acid oxidation [27]. In a later population-based prospective study of 2103 individuals, with a mean follow-up of six years, palmitoylcarnitine was included in a panel of acylcarnitines that demonstrated an AUC of 0.89 for the prediction of incident T2DM [28]. Palmitoylcarnitine was also associated with T2DM in a study of 2519 patients with coronary artery disease with a median follow-up of 7.7 years, and a three-metabolite panel that also contained trimethyllysine and γ-butyrobetaine was proposed for predicting the long-term risk of T2DM, based on the susceptibility to this disease of patients with dysfunctional fatty acid metabolism [29]. An in-depth study of the mechanism underlying this relationship described palmitoylcarnitine as a useful biomarker of excessive fatty acid oxidation, which leads to tissue lipid accumulation and ultimately insulin resistance, finding elevated concentrations of this metabolite in patients with T2DM during the insulin clamp at fasting [30]. Finally, a recent meta-analysis of 46 prospective metabolomic studies on T2DM proposed a panel of 10 metabolites, including palmitoylcarnitine, for this disease [31].

Biliverdin is a dark green bile pigment product of heme catabolism, which can be converted to bilirubin by the catalytic reaction of biliverdin reductase. In mouse cells, deletion of this enzyme using CRISPR-Cas9 technology produced lipid accumulation and oxidative stress that led to non-alcoholic fatty liver disease, which is strongly associated with insulin resistance and diabetes [32]. Elevated concentrations of biliverdin may, therefore, reflect a reduction in bilirubin concentrations. Serum concentrations of bilirubin are inversely related to the development of T2DM, because it protects against insulin resistance by improving visceral obesity and adipose tissue inflammation [33]. Thus, the prevalence of T2DM is markedly lower among patients with Gilbert syndrome, characterized by congenital hyperbilirubinemia [34]. It has also been observed that the administration of biliverdin, which showed lower concentrations in the present patients with T3cDM, ameliorates pancreatic inflammation and has been proposed, alongside methylene chloride, as a possible therapeutic approach to pancreatitis [35]. Hence, the protective role attributed to hemeoxygenase-1 in gastrointestinal diseases may not be direct but rather the result of a cytoprotective and anti-inflammatory response against oxidative stress produced by its upregulation and the production of CO and biliverdin [36].

Finally, the last compound that appeared differentially expressed between the study groups was 7α-hydroxy-3-oxo-4-cholestenoic acid (7-HOCA). This metabolite of cholesterol, which occurs naturally in human blood [37], is formed extrahepatically and has been observed to accumulate in human subdural hematomas in correlation with the time from bleeding to sample collection, and it has also been proposed as a diagnostic marker for dysfunctional blood–brain barrier [38]. This metabolite is taken up by the liver, where it is further oxidized into bile acids, mainly chenodeoxycholic acid [39], which have been shown to acutely enhance insulin secretion [40], explaining the accumulation of this precursor in our T2DM group.

## 4. Materials and Methods

### 4.1. Sample Collection

To avoid any sex-related separation of the groups, we studied blood samples from 21 males with T3cDM (T3cDM group) and 19 males with T2DM (T2DM group) (Table 2). Blood samples were drawn in hospital between 8 a.m. and 9 a.m. from fasting patients with T3cDM. All patients signed their informed consent to participation after receiving written information about the study, which was approved by the ethics committee of the hospital (ethical approval number: 1269-M1-19) and followed the principles of the Helsinki Declaration. Samples were collected in BD Vacutainer SSTII Advance tubes (Becton Dickinson, Franklin Lakes, NJ, USA) containing silica to activate clotting of the specimen. After centrifugation for 10 min at 2450 rpm, the supernatant was aspirated and stored at −80 °C until analysis. Serum samples from T2DM patients were supplied by the Biobank of the Andalusian Public Health System and were obtained and treated in the same manner as the samples from T3cDM.

### 4.2. Metabolite Extraction

All samples in this study were maintained at 4 °C during their handling. Protein precipitation was achieved by using acetonitrile (AcN) (1:8 sample/AcN) and shaking for 2 min, followed by centrifugation at 15,200 rpm for 10 min at 4 °C. Next, the supernatants were moved to HPLC vials and dried out in a GeneVac HT-8 evaporator (Savant, Holbrook, NY, USA). The remaining material was dissolved in a solution of AcN/water [50:50] with 0.1% formic acid and then shaken for 1 min.

### 4.3. LC-HRMS Analysis

Samples were analyzed by means of an LC-HRMS platform integrated by an Agilent 1290 LC system coupled to a Q-TOF 5600 (Triple Quadrupole Time-of-Flight) mass spectrometer (AB SCIEX, Concord, ON, Canada) using electrospray ionization in positive mode. Chromatographic stage was achieved with an Atlantis T3 HPLC column (C18: 2.1 mm × 150 mm, 3 μm) (Waters Corporation, Milford, MA, USA) kept at 25 °C and considering an injection volume of 5 μL of sample. The composition of the mobile phase was as follows: 0.1% formic acid-90:10 water/AcN (eluent A) and 0.1% formic acid-90:10 AcN/water (eluent B). The elution of analytes was accomplished applying the following elution gradient: 0.00–0.50 min 1% eluent B, 0.50–11.00 min 99% eluent B, 11.00–15.50 min 99% eluent B, 15.50–15.60 min 1% eluent B, and 15.60–20.00 min 1% eluent B. The flow was set at 300 µL/min. The Q-TOF 5600 was configured in an information-dependent acquisition (IDA) mode that enabled the simultaneous acquisition of full-scan HRMS and MS/MS data. Ion source parameters and IDA conditions were: gas source 1:50.00; gas source 2:50.00; curtain gas: 45.00; temperature: 500.00 °C; ionspray voltage floating: 4500.00; TOF masses: Min = 80.0000 Da Max = 1600.0000 Da; accumulation time: 0.2500 s, IDA accumulation time: 0.1000 s.

In order to ensure the accurate determination of exact mass, a periodic mass calibration was carried out every 10 injections with a standard solution. The injection sequence of samples included OS samples and quality control (QC) samples every 10 injections. QC samples were drawn from a pool of equal aliquots of all serum samples collected in this study and used to assess the analytical drift of the system. OS samples were used to filter any contaminants from the organic solvents or extraction procedure and to check the effect of carryover contamination.

### 4.4. Data Set Creation

PeakView software (version 1.0 with Formula Finder plug-in version 1.0, AB SCIEX, Concord, ON, Canada) was applied to check the retention time (RT) and mass/charge (*m*/*z*) drift of the experiment. MarkerView software (version 1.2.1, AB SCIEX, Concord, ON, Canada) was selected for raw LC-HRMS data processing. This software enabled peak detection, alignment, data filtering, *m*/*z* and RT determination and ion peak integration for each sample. Calculations were performed in the RT range of 1–17.2 min, and the peak intensity threshold was set at 60 cps. The tolerances of *m*/*z* and RT alignment values were set at 0.10 min and 10 ppm, respectively. Only *m*/*z* values that appeared in at least 10 samples from the study group were included in the data matrix, producing a matrix of 3998 features defined by RT and *m*/*z* values. Next, monoisotopic peaks (886 features) alone were considered to reduce the mass redundancy and enhance the selection of true molecular features. Variables were removed from the data matrix if their reproducibility was inadequate (relative standard deviation [RSD] >30% in QC samples) or if they were observed in <50% of QC samples. As a result, 350 features were removed. Then, mass signals differentially expressed by the OS and case study samples (T3cDM and T2DM) were identified by applying an additional filtering procedure., calculating the mean value of each feature in blank and QC samples (pool of all extracts) and applying a *t*-test. All features with a *p*-value <0.05 and mean blank/mean QC ratio >1.5 were removed from the data matrix (257 features). Finally, a filtered data matrix comprising 278 features was considered for further analysis. The subsequent phases were performed using Metaboanalyst 3.0 Web Server.

### 4.5. Data Pre-Treatment

Normalization was performed using a QC sample and probabilistic quotient normalization, mean centering scaling, and log transformation to transform the data matrix into a more Gaussian-type distribution.

### 4.6. Analytical Validation and Outlier Detection

The relative distribution of QC samples and study samples on a PCA plot score was used to identify the drift of the analytical system. Separately, one outlier in T3CDM and two in T2DM were recognized from Hotelling T2 ellipses in a partial least squares-discriminant analysis (PLS-DA), regardless of the approach used to adjust the standardization method. The removal of outliers did not produce a rise in R2 or Q2 figures (data not shown).

The antibiotic roxithromycin was employed as the analytical standard. All studied samples were spiked after reconstitution of dry residues with 5 uL of a stock solution of roxithromycin to yield a final concentration of 150 ng/mL. This compound has multiple functional groups in its structure and provides a very specific fragmentation pattern, making it a good candidate to identify the stability of the analytical system. Furthermore, it is rarely used in clinical therapy, minimizing any potential inference with endogenous content. In terms of RT and accurate mass values, roxithromycin displayed an excellent reproducibility across the whole set of samples. The RT was 6.37 ± 0.02 min, and the accurate mass was 837.5303 Da ±2ppm. The mean roxithromycin peak area was 1,733,454.78 ± 398,032.56 cps. The CV was 22.96%.

### 4.7. Statistical Analysis

The normal distribution of the data matrix was checked with the Shapiro test. The Student’s *t*-test and Wilcox tests were used to explore significant differences in features between T3cDM and T2DM groups. The significance threshold was set at a *p*-value <0.05, including Benjamini–Hochberg false discovery rate (FDR) correction. Then, multivariate analysis (PCA&PLS-DA) was applied to identify the most discriminant variables between the groups. The feature selection criteria were finally set at an FDR-corrected *p*-value < 0.05 and fold change (T2DM/T3cDM) ranging between <0.6 and >1.5.

### 4.8. Biomarker Identification

The estimation of molecular formula from experimental data (accurate mass, isotopic profile and fragmentation patterns) was done with PeakView software (version 1.0 with Formula Finder plug-in version 1.0, AB SCIEX, Concord, ON, Canada). Next, compound databases (Metlin, Human Metabolome Database, Lipid Maps, PubChem, ChemSpider) and spectral databases (MassBank, NIST2014) were interrogated with the experimental data of each candidate for structural annotation. Based on these criteria, all annotations reported in our study should be taken as tentative. When possible, tentative annotations were confirmed using authentic standards.

### 4.9. Metabolite Evaluation

The AUC was used to assess the predictive potential of the selected candidate biomarkers, either individually or in combination.

## 5. Conclusions

In this study, untargeted LC-HRMS metabolomics proved useful to discover novel biomarkers that differentiate between T3cDM and T2DM. We propose a 13-metabolite panel that classified each type of patient with high accuracy. Sampling procedures and conditions were designed to minimize possible biases and applied in the most homogeneous manner possible. Nevertheless, it can be highly challenging to detect and eliminate all sampling biases, and this represents a potential limitation. A further limitation of this type of study is the difficulty to find replication cohorts to serve as a separate validation set. For these reasons, additional investigation is required in larger samples to identify the ability of these diagnostic markers for T3cDM and T2DM alongside new validation studies, in order to establish the potential clinical relevance of these metabolic biomarkers and deliver the most appropriate treatment to these patients.

## Figures and Tables

**Figure 1 metabolites-10-00423-f001:**
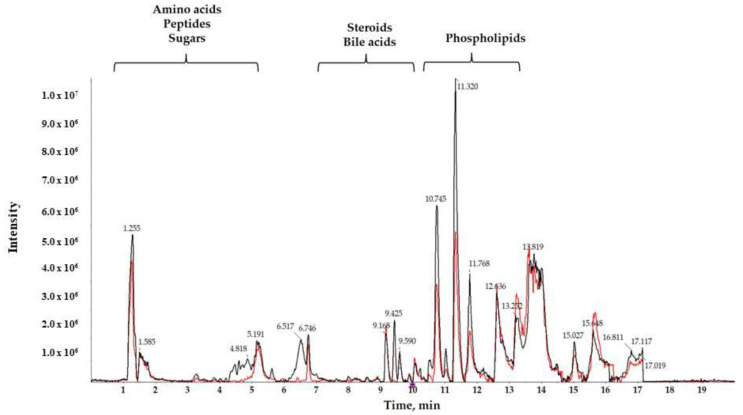
Characteristic liquid chromatography coupled to high resolution spectrometry (LC-HRMS) total ion chromatogram (TIC) of serum samples from T2DM (black) and T3cDM (red).

**Figure 2 metabolites-10-00423-f002:**
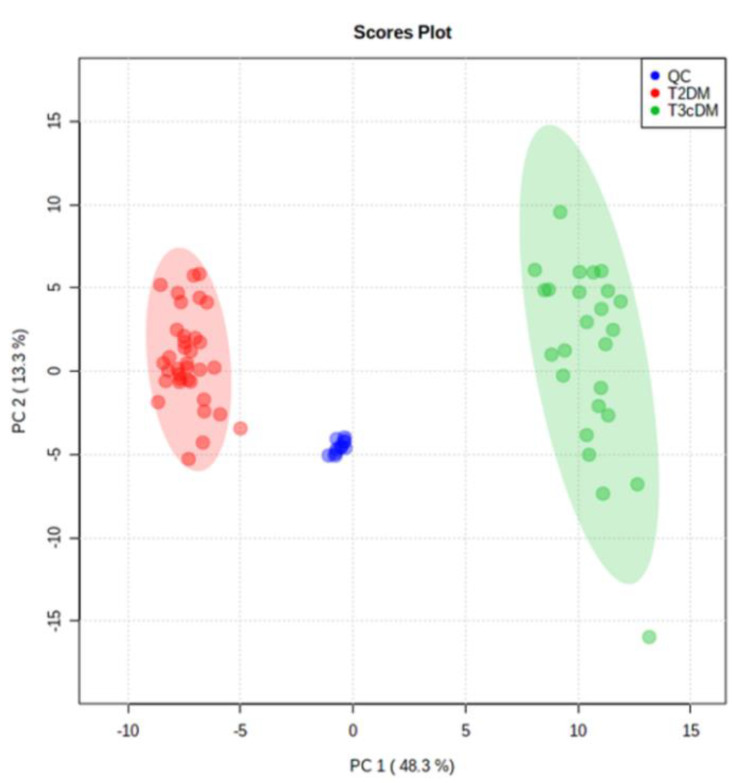
Principal component analysis score plot of TD2DM (red), T3cDM (green), and quality control (QC) samples (blue).

**Figure 3 metabolites-10-00423-f003:**
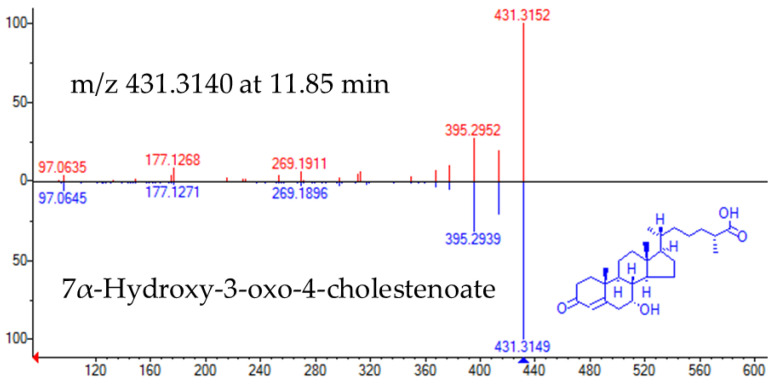
Head to tail representation of fragmentation spectra of *m*/*z* 431.3140 in a biological sample (red trace) and NIST data base spectra for 7α-Hydroxy-3-oxo-4-cholestenoate (pink trace).

**Figure 4 metabolites-10-00423-f004:**
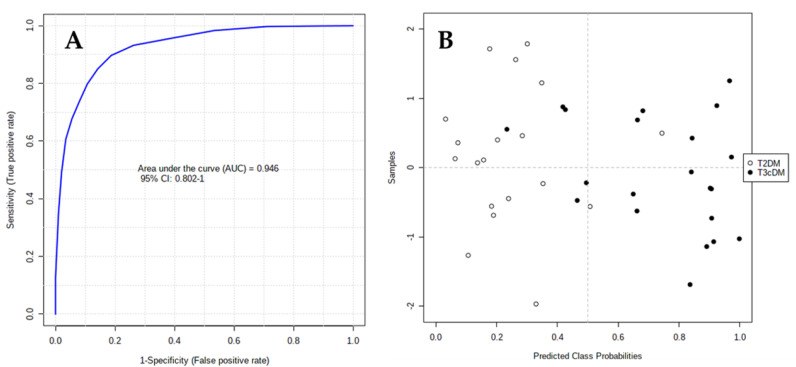
ROC curve plot for the final model obtained by multivariate combination of individual biomarkers: (**A**) ROC curve plot was created from the averaged results of 100 cross-validations. As an outcome from averaged cross-validations, it possible to predict the classification of each patient sample; (**B**) as a consequence of the balanced subsampling approach incorporated in the algorithm, the classification limit is x = 0.5 (dotted line).

**Table 1 metabolites-10-00423-t001:** Detailed information on the potential biomarkers to distinguish between T3cDM and T2DM.

m/z	RT	Adduct	MF	ppm	FDR	FC	AUC	Tentative Identification
188.0716	3.3	[M+H-NH3]+	C11H9NO2	−2	7.86 × 10^−3^	2.9319	0.93	L-Tryptophan
400.3408	9.9	[M+H]+	C23H45NO4	3	2.75 × 10^−2^	2.62	0.81	Palmitoylcarnitine
431.3140	11.85	[M+H+	C27H42O4	4	1.39 × 10^−2^	3.5449	0.80	7-HOCA
526.2931	10.6	[M+H]+	C27H44NO7P	1	7.86 × 10^−3^	2.3703	0.71	LysoPE(22:6)
583.255	8.2	[M+H]+	C33H34N4O6	0	2.75 × 10^−2^	2.9319	0.82	Biliverdin

RT: Retention time in min; MF: Molecular formula; ppm: mass error in parts per million; FDR: false discovery rate; AUC: area under receiver-operating characteristic curves; FC: Fold (T2DM/T3cDM); 7-HOCA: 7α-hydroxy-3-oxo-4-cholestenoic acid; LysoPE: lysophosphatidylethanolamine.

**Table 2 metabolites-10-00423-t002:** Patient characteristics.

	T3cDM	T2DM
n	21	19
Age, mean years (±SD)	58.42 (±9.57)	55.96 (±6.16)
Sex		
Male	21	19
Female	0	0
Caucasian	21	19
Stage		
A	0	-
B	0	-
C	21	-
Disease duration, years	5.36	7.14
HbA1c, %	7.82	7.69
BMI, mean (±SD)	21.4 (±6.61)	24.3 (±3.54)
Fasting glucose, mg/dL	144.36	138.42
Insulin treatment	21	19

T3cDM = patients with pancreatogenic diabetes mellitus; T2DM = patients with type 2 diabetes mellitus; HbA1c = Hemoglobin A1c; BMI = Body mass index.

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
