# Peer review of "Novel Biomarkers to Distinguish between Type 3c and Type 2 Diabetes Mellitus by Untargeted Metabolomics"

_metabolites, 2020, doi:10.3390/metabo10110423_

Round 1

Reviewer 1 Report

Congratulations on a well-performed revision. The manuscript has been greatly improved. I would suggest that the authors mention a few limitations of the study in the discussion (see below). This can be easily accomplished in a minor revision.

Point 1:

Sampling is now well explained. However, as biases may be extremely difficult to find and eliminate, I still suggest that this is mentioned, together with the precautions taken, as a limitation of the study.

Point 2:

Parameters presented in Table 3 should be statistically tested for differences between groups. If there is a difference, then analysis of metabolite levels should be adjusted for these factors.

Point 3:

Thanks for clarifying this. In the previous version I misunderstood it as filtering based on differences between experimental groups. The description is now massively improved.

Point 4:

My critique here is largely resolved as Point 3 has been resolved. I understand the problem of finding replications cohorts. Please mention this as a limitations of the study.

Point 5:

This has been resolved.

Author Response

Comments and Suggestions for Authors:

Congratulations on a well-performed revision. The manuscript has been greatly improved. I would suggest that the authors mention a few limitations of the study in the discussion (see below). This can be easily accomplished in a minor revision.

Point 1: Sampling is now well explained. However, as biases may be extremely difficult to find and eliminate, I still suggest that this is mentioned, together with the precautions taken, as a limitation of the study.

Response 1: In accordance with the reviewer we have included the following statement in the Conclusions section:

Sampling procedures and conditions were designed to minimize possible biases and applied in the most homogeneous manner possible. Nevertheless, it can be highly challenging to detect and eliminate all sampling biases, and this represents a potential limitation.” (Page 9; line 318 to 321).

Point 2: Parameters presented in Table 3 should be statistically tested for differences between groups. If there is a difference, then analysis of metabolite levels should be adjusted for these factors.

Response 2: All of the parameters were statistically tested for between-group differences, and no significant differences were found.  

Point 3: Thanks for clarifying this. In the previous version I misunderstood it as filtering based on differences between experimental groups. The description is now massively improved.

Response 3: We are grateful for this comment.

Point 4: My critique here is largely resolved as Point 3 has been resolved. I understand the problem of finding replications cohorts. Please mention this as a limitations of the study.

Response 4: This has been done, adding the following sentence in the Conclusions section:

“A further limitation of this type of study is the difficulty to find replication cohorts to serve as a separate validation set.” (Page 9 and 10; line 321 to 322).

Point 5: This has been resolved.

Reviewer 2 Report

Thank you for incorporating the suggestions. I have no further comments. 

Author Response

Comments and Suggestions for Authors: Thank you for incorporating the suggestions. I have no further comments.  

Response 1: We are grateful to this reviewer for his help in improving the manuscript.